# Recent Developments of High-Pressure Spark Plasma Sintering: An Overview of Current Applications, Challenges and Future Directions

**DOI:** 10.3390/ma16030997

**Published:** 2023-01-21

**Authors:** Yann Le Godec, Sylvie Le Floch

**Affiliations:** 1Institut de Minéralogie, de Physique des Matériaux et de Cosmochimie (IMPMC), Sorbonne Université, UMR CNRS 7590, Muséum National d’Histoire Naturelle, IRD UMR 206, 75005 Paris, France; 2Institut Lumière Matière, UMR5306 Université Lyon 1-CNRS, Université de Lyon, CEDEX, 69622 Villeurbanne, France

**Keywords:** spark plasma sintering, high pressure, HP-SPS

## Abstract

Spark plasma sintering (SPS), also called pulsed electric current sintering (PECS) or field-assisted sintering technique (FAST) is a technique for sintering powder under moderate uniaxial pressure (max. 0.15 GPa) and high temperature (up to 2500 °C). It has been widely used over the last few years as it can achieve full densification of ceramic or metal powders with lower sintering temperature and shorter processing time compared to conventional processes, opening up new possibilities for nanomaterials densification. More recently, new frontiers of opportunities are emerging by coupling SPS with high pressure (up to ~10 GPa). A vast exciting field of academic research is now using high-pressure SPS (HP-SPS) in order to play with various parameters of sintering, like grain growth, structural stability and chemical reactivity, allowing the full densification of metastable or hard-to-sinter materials. This review summarizes the various benefits of HP-SPS for the sintering of many classes of advanced functional materials. It presents the latest research findings on various HP-SPS technologies with particular emphasis on their associated metrologies and their main outstanding results obtained. Finally, in the last section, this review lists some perspectives regarding the current challenges and future directions in which the HP-SPS field may have great breakthroughs in the coming years.

## 1. Introduction

Over the past twenty years, the spark plasma sintering technique (SPS) has become a common tool in advanced materials research, including metals, ceramics, polymers and composites. Spark plasma sintering (SPS) is a sintering technique under moderate pressure (up to 150 MPa) and at high temperature (up to 2500 °C). It allows the production of bulk materials by densification of powders in shorter times and at lower temperatures compared to conventional processes. The originality of this technique is based on the use of a high-intensity, low-voltage, pulsed current, which flows through the graphite mold containing the powder and eventually through the sample. The heating induced by the Joule effect is therefore very fast (up to 1000 °C/min). The combination of high heating and cooling rates with a uniaxial applied pressure results in fast consolidation of powder at temperatures 200 to 500 °C lower than with conventional sintering [1]. Indeed, SPS is one of the techniques among the fast sintering approaches allowing the production of dense nanostructured materials by tuning the kinetics and the uniaxial pressure to prevent grain coarsening and to favor densification phenomena. The efficiency of the SPS method has been attributed to the sample exposition to a pulsed electric field during the sintering process. Zhang et al. [2] have shown first evidence for the occurrence of a spark discharge during SPS sintering of copper powder or Ti-TiB_2_ mixture. They claim the generation of high-temperature spark plasma in the gaps between the powder particles at the beginning of on-off DC pulse energizing. Fast, efficient sintering of conductive powders would be then achieved under the combined action of spark discharge, Joule heating, electrical diffusion and plastic deformation in the SPS process. Sakka et al. summarize the present status and problems of SPS technology for the sintering of conductive powders [3].

In the case of insulating powders, there is no direct evidence of spark discharge events [4]. Thus, the role of the pulsed current remains not clearly understood. Many papers report analyses of the mechanisms involved during spark plasma sintering of non-conducting ceramics [5,6,7]. The exact nature of the SPS mechanism is therefore still under debate.

Despite this, due to its unique characteristics, SPS is the technique of choice for the densification of hard to sinter materials, such as highly refractory, nanocrystalline and metastable materials. SPS is also used for other applications such as the manufacture of highly porous materials or the production of multi-layer materials, by welding of dissimilar thin-film materials or by sintering powder stacks into laminates [8,9]. The very short duration and relatively low temperature involved in this technique also make it very attractive for densification and preservation of the nanoscale grain size to produce bulk nanostructured ceramics, for which grain growth control is one of the main issues. SPS is a viable and important tool for producing controlled microstructures leading to materials with improved properties. Tokita et al. [10] and Guillon et al. [11] provide overviews of the progress of SPS technology and its industrial products application.

At the same time, regardless of this research on SPS sintering, high-pressure science has recently become a booming field of multidisciplinary research. As a thermodynamic intensive variable, pressure can induce fascinating changes in chemical, structural, electronic, magnetic, optical and elastic properties of materials, resulting in numerous spectacular phenomena [12]. For example, pressure can profoundly modify the nature of materials, in some cases transforming soft materials into superhard ones [13], insulators into conductors [14] (or conversely [15]), ferromagnets into superconductors [16,17], crystalline structures into amorphous ones [18], or even chemically very reactive compounds into fully inert materials. High-pressure processing of nanomaterials also opens up a wide range of research, resulting in new materials [19]. This is why, during recent decades, the impact of high-pressure science has expanded rapidly to cover various scientific and technological fields, from planetology to physics and chemistry of materials through to biology and the food industry.

For instance, in physics, pressure makes it possible to explore new phenomena, thus providing critical tests for theories linking advanced properties and crystal structure/composition. A paradigmatic example concerns the recent discoveries of high-pressure materials with record superconducting temperatures, for example hydrogen sulphide (Tc = 203 K) [20] or a new hydrogen-rich carbon material [21] (Tc = 288 K, i.e., the average surface temperature of the Earth!). These studies have generated enormous enthusiasm within the superconductivity research community, stimulating new important research programs.

In planetology, pressure is a crucial study parameter since the vast majority of the materials composing these planets are found under high pressure (for example, more than 99% of the Earth’s mass is at pressure higher than 1 GPa). High-pressure research is therefore essential to understand the structure of the planets and the internal processes (magnetism, heat transfer and volcanism, etc.) involved.

Finally, in chemistry and materials science, high-pressure studies can allow the synthesis of new compounds with exceptional properties, which are usually impossible to synthesize under the usual conditions of atmospheric pressure. Indeed, following historical studies on the high-pressure synthesis of diamond and then of cubic boron nitride in the 20th century, the most recent discoveries concern a wide range of new strategic materials which can have exceptional mechanical, energetic, magnetic, optical or thermoelectric properties, making it possible to consider important technological applications [22].

Until very recently, the two scientific fields—high-pressure science and spark plasma sintering—continued their separate developments without combining the two technologies. However, new frontiers of opportunities are now emerging by coupling SPS with large volume (i.e., at least for samples greater than several cubic millimeters) high-pressure technology (up to ~10 GPa), allowing the full densification of hard-to-sinter materials.

In this review, the various benefits of the HP-SPS field for sintering many classes of advanced functional materials are first conceptually presented (Section 2). Then we focus on the recent progress of various HP-SPS technologies with particular emphasis on their associated metrologies and their main outstanding results (Section 3). In the fourth section, this review highlights some of the most important perspectives regarding the current challenges and future directions in which the HP-SPS field may have great breakthroughs in the coming years. Finally, a general conclusion summarizing this new HP-SPS field ends this review.

## 2. Why Couple SPS with High Pressure?

Various “external” parameters are generally associated with the conventional SPS process and are used to optimize sintering according to the “internal” parameters, as for instance the nature of the sample (crystal structure and chemical composition, etc.) or the size of its particles. These “external” parameters are basically the current (nature, pulse life-time and accordingly the sample temperature), the pressure applied, the heating rate, the holding time (or dwell time) and then the cooling rate. The influence of each of these parameters has been widely investigated [23] and every new study into a particular scientific case must optimize these parameters to obtain the best possible result. Among all these “external” parameters, pressure plays a central role (even in traditional SPS systems where this pressure is still limited by the mechanical property of the graphite die to ~150 MPa). Several experimental observations and/or conceptual arguments that make high pressure a major asset for densification in sintering are detailed below:The applied pressure has the mechanical effect of reducing the distances between the particles and strengthens the contact between the individual grains. Therefore, the time required for atomic diffusion and mass transfer in the process is reduced when pressure is applied and, therefore, pressure has been shown to drastically decrease the overall sintering time.It has been demonstrated that pressure can greatly reduce the sintering temperature [24]. Pressure actually increases the driving force for densification by modifying diffusion-related mass transport, viscous flow, plastic flow and creep [23]. This effect is crucial, for example to limit the grain growth (which is activated by high temperature). Validation of this important effect has been provided in many recent SPS studies [25,26,27,28]. For example, Figure 1a shows that, with SPS, the growth of alumina crystallites is remarkably inhibited by increasing the applied pressure from 30 MPa to 100 MPa [26]. Similarly, Figure 1b shows the pressure required to obtain nearly fully-dense nanometric zirconia samples (relative density of 95%) using a dwell time of 5 min at the indicated temperature, also with the grain size achieved under the specified conditions. It is demonstrated that, under high pressure, the temperature required to reach 95% of the theoretical density is several hundred degrees below the typical sintering temperatures (1300–1500 °C) required using traditional low-pressure sintering [25].

3.Furthermore, reducing the sintering temperature with pressure can be crucial for keeping a phase which undergoes a phase transition at the low-pressure sintering temperature. Some examples can be found in the literature, for example the densification of alpha-quartz SiO_2_ [29] or amorphous calcium phosphate [30].4.In the same vein, increasing applied pressure becomes essential to allow sintering at temperatures below the decomposition temperature of the compounds (for example MgB_2_ [31], BP [29], etc.) or to increase the thermal stability of precursors, for example by confining OH, H_2_O or other volatile elements [29].5.Pressure also allows sintering of a metastable phase under ambient conditions in its high-pressure thermodynamic stability domain (for instance, diamond and cubic-boron nitride, etc.) [32].6.It has been demonstrated that pressure enables sintering of refractory materials (which are hard-to-sinter materials because the sintering temperature is usually too high at low pressure), reaching a density of almost 100% (example boride, nitride or carbide) [33,34].7.Pressure can induce a phase transition during the process leading to sintering of a denser phase (e.g., synthesis and sintering of diamond from graphite or c-Si_3_N_4_ from initial α-Si_3_N_4_) [29]. For example, Figure 2 shows the XRD patterns of the SPS-processed silicon carbide at 40 and 80 MPa. At 1900 °C, the high pressure induced a complete phase transition from the initial beta phase to the alpha phase which remains after sintering [35].

8.High pressure can initiate a finer microstructure than the grain size of the starting powder by driving the high-pressure polymorphism of the materials (ex: α-Al_2_O_3_ from γ-Al_2_O_3_ [36] or in TiO_2_ [37]).9.Finally, extrinsically, through particle packing, sliding, fragmentation and deformation, pressure greatly influences particle reorganization, leading to a better homogeneous densification by the destruction of agglomerates in powders, especially in nanopowders. The crucial role of pressure in the particle rearrangements has been shown for example in the SPS process of Si_3_N_4_ ceramic (with LiF) [38] and for the densification of a reactive mixture of ZrO_2_ and Y_2_O_3_ [39].

In short, all the advantages brought by pressure explain the recent development of many new techniques to considerably extend the range of pressure accessible in SPS. We will now present these many studies in Section 3 of this paper.

## 3. Recent Progress of Various HP-SPS Technologies

### 3.1. Basic Configuration of SPS Apparatus

There are a large number of reviews describing in detail the fundamental process (related to parameters on atomic diffusion and kinetic mechanisms, etc.) of the SPS technique [3,10,23,25,40,41,42]. In its basic configuration, the SPS apparatus simultaneously applies a pulsed high DC current along with uniaxial pressure to densify powders or synthesize (and consolidate) new compounds. It is schematically shown in Figure 3.

Basically, the SPS system consists of a hydraulic pressure device (usually 250 kN or equivalent) with a vertical pressurization axis (part A of Figure 3). The pressure is transferred to the reaction chamber (part B of Figure 3). This reaction chamber is usually placed in a vacuum (or a protective atmosphere, e.g., H_2_, N_2_, and argon) and includes two water-cooled steel cylinders (rams), two graphite spacers, a cylindrical die (usually in graphite), where the sample to sinter is placed, and two graphite punches pressing the sample.

A DC generator (part C of Figure 3) is connected to the two steel rams and produces a pulsed direct current flowing through the graphite punches, sintered powder (in appropriate cases) and particularly through the graphite die (i.e., the die serves both as pressure cylinder and external heating elements), leading to very rapid and efficient heating. The direct current is regulated by pulses and the pulses usually have a duration of 3.3 ms. The number of pulses per time unit can be varied but, generally, for commercial SPS, a pulse pattern of 12–2 is recommended by manufacturers, meaning that twelve pulses are applied, followed by a duration of two pulses where the current is not applied (6.6 ms). Heating rates as high as 1000 °C/min are possible and the temperature is monitored at different places of the die using both thermocouple and IR pyrometer.

The simultaneous output of primary pressure, temperature, vacuum level and displacement (shrinkage) is controlled using a computer with appropriate software (part D of Figure 3) which can monitor the hydraulic press, the vacuum system and the DC generator.

In this basic SPS configuration, pressure can be applied and released very fast and, compared to conventional hot pressing sintering, “high pressures” can be reached (up to 150 MPa).

While simple modifications can be made to extend this limit to 500 MPa (see for example [43,44]), the system must be radically modified to be able to go beyond this. Thus, in recent years, many research groups have worked to extend this range well beyond 500 MPa by modifying: (i) the reaction chamber (part B of Figure 3), (ii) the reaction chamber part and the hydraulic pressure device (parts A and B of Figure 3), (iii) the entire SPS system (parts A, B, C and therefore D). In the following, we will explain in detail these various options and give a simple explanation of the different solutions found.

### 3.2. Modification of the Reaction Chamber

To overcome the limit of pressure (150 MPa) imposed by the high temperature fracture strength of the graphite, various modifications of the pressing tool have been implemented. Keeping an outer and an inner graphite die and replacing the two graphite punches with two tungsten carbide (WC) [45] or silicon carbide (SiC) punches [44], 400–500 MPa can be reached. Additionally replacing the inner graphite die with WC [43,46] or SiC [47,48] does not allow 500 MPa to be exceeded on a 10 mm diameter sample. It should also be noted that the operating temperature of the WC tools is generally limited to 700 °C, whereas SiC tools allow heating up to 1300 °C [47].

In order to reach higher pressure, the diameter of the sample can be reduced to 5 mm. Using SiC punches with external and inner graphite dies, pressures as high as 1 GPa are achieved and heating up to 900 °C is reported [25,49]. The HP-SPS device developed by Anselmi-Tamburini et al. is shown in Figure 4. Temperatures were measured using a shielded K-type thermocouple inserted in the lateral wall of the external die. The actual sample temperature was determined through a calibration curve obtained by placing a second thermocouple in the center of the sample.

To apply 1 GPa on a 10 mm diameter sample, Ratzker et al. designed a HP-SPS hybrid SiC-graphite tool, shown in Figure 4B [50]. They used an outer graphite die with an inner SiC die and modified the shape of the SiC punches. This device allows SPS up to 1100 °C, but the heating rates seem to be limited to 50 °C/min up to 800 °C and 12.5 °C/min beyond. They obtained transparent nanostructured alumina discs 10 mm in diameter and about 1.2 mm thick.

To go beyond 1 GPa, we adapted a very high-pressure module, based on a Paris–Edinburgh high-pressure cell (PE cell) [51,52], to standard SPS equipment [48]. In this case, a pellet (7 mm in diameter and 3 mm thick) of pre-compacted powder (at 370 MPa) is inserted into a graphite tube surrounded by a gasket made of fired pyrophyllite. This assembly is placed between two opposed anvils with a pseudo-conoidal profile which reduces the uniaxial stress (Figure 5A). The anvils are made of cobalt-cemented tungsten carbide supported by a steel binding ring.

The pressure is generated by applying load on the anvils. The gasket plastically flows and transfers a quasi-hydrostatic pressure to the sample. A thermoplastic PEEK (polyether ether ketone) supporting ring around the gasket enhances the pressurization efficiency. The calibration of the pressure was previously performed following the measurement of the equation of state of NaCl by in situ neutron diffraction at the ISIS facility [53] (Figure 6A). As the maximum load applied with the Dr. Sinter SPS 825 and the SPS-HPD 25 (FCT) apparatus is 250 kN, we achieve a maximum pressure of 2 GPa with the Paris–Edinburgh module mounted in the SPS chamber. The temperature is measured at the top of the sample with an insulated thermocouple passing through the drilled anvil.

Nevertheless, during sintering cycles, the thermocouples often break. So, for safety reasons, we do not regulate the heating on the temperature measurement, but we gradually increase the electric power. Temperature calibration was previously performed by placing a K-type thermocouple in the center of an alumina sample (Figure 5A and Figure 6B). Heating rates of 100 °C/min are generally used. As the high-pressure module is not water cooled, the heating temperature is limited to 900 °C to preserve the anvils. At 1.5 GPa, 7 mm diameter and 1 mm thick discs of transparent nanostructured alumina have been produced in 6 min at 900 °C.

### 3.3. Modification of the Reaction Chamber and the Hydraulic Pressure Device

To extend the pressure range, modifying the reaction chamber of an SPS apparatus is no longer sufficient. Indeed, the presses generally used in the commercial models of SPS are limited in capacity (generally 50–250 kN). This turns out to be insufficient if the reaction chamber is more complex, especially if a high-pressure conventional system of anvils/gaskets is used. This is why many research teams have recently developed new SPS systems where the reaction chamber and the hydraulic pressure device are modified to reach very high pressure for spark plasma sintering.

Historically, the first occurrence of such a system is described in a patent filed by an American industrial group in 2010 [54]. Indeed, this new HP-SPS technology was developed for industrial production purposes, mainly to commercialize new ultra-hard material cutting elements. Figure 7 shows the working principle of the cubic press where the cubic reaction chamber can be triaxially compressed via six independent electrically-insulated tungsten carbide anvils. The latter are driven by a computer-controlled hydraulic system to apply the same pressure on each side of the sample assembly (Figure 7), assuring a better hydrostaticity of the compression.

The maximum load on each of these six rams is about 10 MN for most industrially available cubic presses, even if this information is not clearly mentioned in the patent (or the exact dimensions of the six anvils or the sample volume, cf. below). Electrical power connections are set on the top and the bottom anvils, where a controlled pulsed DC current (with an amperage in the range of 1 to 10 kA and a voltage of 5 to 10 Volts) is applied via a current ring to the graphite heater in the reaction chamber for generating high temperature.

Indeed, the cubic cell assembly developed for HP-SPS is illustrated in Figure 7 and consists of a gasket material (to thermally insulate the reaction chamber from WC anvils), two steel electrodes filled with a “synthetic gasket material” inside (to prevent dissipation of heat through the top and bottom WC anvils), two metallic disks (to increase the electrical contact area), one baffle to prevent the two discs from sliding against each other under high pressure and the graphite heater. In this latter, the sample to sinter is placed into a refractory metallic capsule (usually in niobium) surrounded by a salt capsule (cf. Figure 7). This metallic capsule has a rather particular shape in two parts because it corresponds to the intended applications: the powder to sinter (usually diamond or polycrystalline cubic boron nitride) is placed on a substrate such as cemented tungsten carbide. The maximum pressures (between 5 and 7 GPa) and temperatures (2000 °C) are similar to conventional cubic press devices.

Unfortunately, no final sintered sample size is given in the patent. That said, given the accessible pressure range mentioned and knowing the conventional assemblies of cubic presses that are actually the main workhorse of the synthetic diamond industry (especially in China), we can estimate that the cubic gasket (part 3 of Figure 7) is of the order of 30 × 30 × 30 mm^3^. Hence, that would mean a sample sintered by this device of the order of 5 mm in diameter for 3 or 4 mm in height. No indication is given either as to the measurement of pressure and temperature in this system and no scientific publication relating to this device has been published since this patent in 2010. Nevertheless, it seems that this first HP-SPS (>5 GPa) has been used in industry for the sintering of ultra-hard materials.

After this first pioneering study, an academic group from the University of Krakow, Poland, developed in 2016 a new HP-SPS device using a Bridgman-type toroidal large-volume apparatus [55]. Although, as in the cubic press mentioned in the previous system [54], the pressure on the sample is obtained by reducing the volume of a gasket compressed between several anvils, there is a fundamental difference between the two systems. In the cubic press, there are six anvils acting simultaneously and during the compression, the gasket extrusion is severely limited on all sides, while the only two Bridgman-type toroidal anvils permit free extrusion in the equatorial plane of the anvils and pressure is only retained within the sample chamber thanks to the friction in the gasket. As shown in Figure 8, the toroidal shape of the anvils has a double effect: it makes it possible to reduce the lateral extrusion of the central part of the gasket and greatly reduces the shear stresses in the anvils, which leads to an increase in the maximum achievable pressures [56].

Thus, the HP-SPS device (Figure 9 shows a photograph of the system in Krakow) consists of a high-capacity hydraulic press (about 25 MN) compressing two opposed toroidal anvils and a kHz direct pulsed current generator with a maximum current of 15 kA, both controlled by a computer system. Typically, the reaction chamber (Figure 8) consists of a high-pressure transmitting gasket (in ceramic, usually limestone or pyrophyllite) in which a graphite furnace is inserted. The sample (pre-pressed at 100 MPa by uniaxial pressing into pellets with 15 mm diameter and 5 mm thickness) is loaded at the center. Sleeves and spacers are typically used to improve hydrostaticity and thermal homogeneity during the SPS process. The maximum pressure that can be achieved in this HP-SPS device is 8 GPa and the operating sintering temperature range is 20–2400 °C.

Prior pressure and temperature calibrations are required due to the lack of possibility to directly measure the imposed (P,T) sample conditions during sintering processes. With regard to pressure metrology, it is no longer possible, as in the case of a conventional SPS device, to determine the pressure on the sample by a simple measurement of the measurable primary pressure on the ram. Indeed, in the HP-SPS device, the gasket deforms plastically during compression and therefore the relationship of proportionality between sample pressure and primary load is no longer possible. Usually, the apparatus is calibrated by establishing the relationship between the sample pressure and the load. For that, since the gap between the anvils remains sufficiently large for leading in a large number of measuring wires, it is possible to determine at room temperature the correspondence of primary load and resistance transition or the metallization pressure transition point of known compounds such as Bi, Cs, CdTe, Ba, Sn, ZnSe, ZnS and GaAs, etc. Thus, one obtains a calibration curve, which can be used for HP-SPS experiments.

Finally, for this HP-SPS device, temperature measurement is not possible during the process. Prior calibration is also necessary. For this, a special gasket was prepared with a thermocouple, shown in Figure 8a. Thus, a calibration of the temperature of the sample was carried out, where the dependence of this temperature on the characteristics (power) of injected current was thoroughly determined.

This new HP-SPS device was used to sinter ZrC-20 wt%Mo and ZrC-20 wt%TiC composites with a pressure up to 7.8 GPa and temperatures of 1550 °C and 1950 °C. The mechanical performances of the composites were compared to those of an ordinary ZrC produced by the same method. Both composites outclassed pure ZrC, showing quite superior hardness and indentation fracture toughness [55], highlighting the major interest of the HP-SPS technique. In a second study, similar results were observed for Ti–Al–Si alloys [57].

More recently, this HP-SPS device was used to conduct a very interesting comparison between the sintering of diamond composite with Ti + 2B mixture, first using a “classical” high-pressure–high-temperature (HP–HT) press and secondly using this SPS-HP device under the same (P,T) conditions (at 8.0 ± 0.2 GPa and at the temperature of 1650 ± 50 °C during 20 s of dwell time) [58]. During this study, the graphitization process, which is a very important factor when sintering diamond powders, is found to be much more important during sintering with the HP–HT device than with the HP-SPS one, as proven by transmission electron microscopy analysis (cf. Figure 10). No clear explanation of these differences is provided in the reference [58] but these observations show that HP-SPS device offers a very promising prospect for obtaining diamond-based materials with improved properties.

Recently, another HP-SPS device has also been developed at ICMCB laboratory (Bordeaux, France) [59]. The originality of this development is the use of a belt-type press as a device for generating high pressures, as shown in Figure 11.

Since its introduction by Tracy Hall in the mid-1950s [60], the “belt” apparatus is an HP–HT device widely used in the academic and industrial world to achieve extreme (P,T) conditions. The principle is simple: the applied load is divided between the central compression part, where the sample is located, and the external support (the “belt”), used to contain the pressure. Thus, the central die and the pistons are held in compression by support rings, so as to confer a residual stress to counter the very high pressures generated during a high compression. In the HP-SPS belt system, the pressure device consists of a chamber inside a tungsten carbide die with an inner diameter of 32 mm and a cylindrical cavity 29 mm high (cf. green part in Figure 11). Great care has been taken in the design of the die assembly to achieve the optimum tangential stress–strain response within the die during load application. The different rings of the system are made of various materials in order to trap the zone of plastic deformation and minimize the failure modes as fatigue and fracture [59].

Figure 11 shows the sample chamber that consists of fired pyrophyllite cylinder and crude pyrophyllite conical gaskets. The sample is placed into a graphite heater tube and (sometimes) a pressure transmitting medium inside this ~25 cm^3^ free volume pyrophyllite tube. The two molybdenum discs allow better electrical contact while the mica ring visible in Figure 11 helps to control the path of the electrical current to avoid overheating the tungsten carbide die.

The sample has a diameter of 17 mm if it is in direct contact with the graphite furnace. However, it is also possible to sinter samples with a “classical” diameter of 10 mm, by interposing a hexagonal boron nitride (h-BN) capsule to obtain a sample size comparable to conventional 10 mm diameter SPS devices.

The belt press is a blind device without in situ pressure and temperature control and therefore these thermodynamic parameters must be calibrated before any sintering experiment. That is why the pressure was first calibrated using the pressure-induced transition of electrical resistivity of pressure sensors such as bismuth (Bi), thallium (Tl) and barium (Ba) at room temperature. The temperature calibration curve as a function of the delivered electrical power was performed by detecting metal melting temperatures (indium (In), lead (Pb), silver (Ag) and nickel (Ni)) under pressure. As a result, this HP-SPS apparatus allows pressures as high as 6 GPa to be reached in the temperatures range from room temperature to 1800 °C with 10 MN load of the hydraulic system [59].

During recent years, this new device was used to demonstrate that the direct sintering of γ-Al_2_O_3_ into α-Al_2_O_3_ occurs at a much lower temperature than in a conventional HP–HT system [59,61]. The HP-SPS clearly allowed a higher density of bulk α-Al_2_O_3_ to be obtained, as shown in Figure 12.

This HP-SPS has also been used for assembly/consolidation of drills (used for oil and gas exploration drilling) under high pressure by joining tables of WC/Co substrate by polycrystalline diamond compact [59]. Here again, sintering is obtained at a lower temperature than in conventional HP–HT configurations allowing production of shear cutters for drilling bits in a very economical way. Another example concerned the sintering of dense magnesium diboride MgB_2_ where the pressure increase (in the range of 1.7–5 GPa) stabilized the phase above its low-pressure decomposition temperature, thus avoiding the formation of non-superconducting phases such as MgB_4_. The high sintering temperature at high pressure yielded high mechanical hardness in MgB_2_ (1488 HV) and promoted sintering up to a relative density of 100% with a homogeneous fine-grained microstructure required to obtain high current density [31].

In a more recent study, this HP-SPS device allowed the densification of high-pressure phases such as diamond at very unexpected (P,T) domains [32]. Indeed, binderless microcrystalline diamond powders (0.75–1.25 and 8–12 µm) have been sintered at 4–5 GPa and temperatures between 1300 and 1400 °C, making HP-SPS a promising tool for the sintering of various ultrahard or hard materials such as c-BN or other borides, carbides or composites.

### 3.4. Modification of the Entire SPS System

We recently developed a compact very high-pressure SPS by coupling a homemade high intensity pulsed current generator to a Paris–Edinburgh press [52]. This tabletop HP-SPS device weighs less than 60 kg and fits on a surface of 50 cm by 30 cm (Figure 13). Compared to conventional SPS devices, it can be easily installed on any laboratory bench or even in a glove box.

The HP-SPS cell placed inside the Paris–Edinburgh press (2500 kN) is the same as the one we used in the HP-SPS module (Figure 5). Here, the anvils are water-cooled and temperatures as high as 2000 °C can be achieved. These WC-Co anvils are connected to the pulse generator with two large flat copper electrodes. The DC current of a 10 V-1000 A generator is transformed in 2.5 V-600 A pulsed current by four electrolytic low resistance and low inductance capacitors mounted in series. Three power MOFSETs handle the switching of high currents. A programming interface controls the switching times to produce different pulse patterns (Figure 14). The minimum width of one pulse is less than 1 ms. A Hall sensor measures the intensity of the current pulses. The heating rate is adjustable from 1 to 500 °C/min. Different geometries of anvils can be used depending on the desired pressure. The diameter of the sample must be decreased for higher pressures: 7 mm up to 5 GPa and 1 mm up to 10 GPa. The height of the final sintered sample is about 1 to 2 mm whatever the geometry used. The current density through the graphite mold can reach 2000 A/cm^2^. The temperature can be measured in the upper part of the sample with a thermocouple. Two displacement sensors, attached to anvils, allow registration of the shrinkage curve of the sample during the pressure-temperature cycles.

Under 1 GPa, an initial step of densification of the anatase powder is evidenced at temperatures as low as 250 °C, in Figure 15. The shrinkage observed at 400 °C corresponds to the anatase to rutile phase transition, which is lower under pressure. A 98% dense nanostructured ceramic is obtained by following the short sintering cycle up to 650 °C, shown in Figure 15 [48].

Polycrystalline diamond compacts (PDC) are widely used as hard materials for drilling or cutting tools. PDC are fabricated by sintering micrometric powders of diamond using a binder (mostly cobalt). To protect the diamond phase from graphitization, this sintering must be carried out at least under 5 GPa as heating up to 1500 °C is necessary. The amount of binder and the grain size of diamonds have an important impact on the mechanical properties and thermal stability of the PDC. In general, the hardness of PDC is about 70 GPa and its thermal stability is 400–500 °C, which limits the life of tools exposed to frictional heating when cutting or drilling hard materials. Since 2003, binderless diamond compacts with exceptional properties (ultra-hardness, high thermal resistance and high transparency) are fabricated from the conversion of graphite under pressure higher than 15 GPa and temperature higher than 2100 °C. The outstanding nature of this diamond compact is linked to its nano-structure and the absence of binder between the nano-sized grains. Despite the drastic manufacturing conditions, Sumitomo Electric (SEI) has been marketing this nano-polycrystalline diamond (NPD) since 2011 [62]. A great deal of research is currently underway to lower the pressure conditions for its manufacture, with a view to extending the industrial production capacity of this exceptional material. Whatever the source of carbon, its complete conversion into diamond requires a pressure above 15 GPa. The most relevant route is the sintering of nanometric diamond powders by HP-SPS, even if these nanopowders are known to be very difficult to sinter [63]. Under 5 GPa and rapid heating (500 °C/min) up to 1900 °C, we were able to densify a 20 nm grain size diamond powder, forming a sintered pellet of 2 mm diameter, without grain growth or graphitization (Figure 16).

## 4. Prospects for Technological Breakthroughs

The HP-SPS domain is currently a rapidly expanding field with devices that exist in several laboratories in Europe. Many studies have already been carried out using these HP-SPS devices and have demonstrated the relevance of the field for sintering materials that are usually difficult to sinter. These HP-SPS devices, few in number today, are constantly improving and are currently being developed in several universities all over the world. In addition, several technological developments are currently taking place to significantly expand the possibilities of these devices in several directions. In view of this paper, we will mention only three main developments currently underway that will considerably extend the possibilities of HP-SPS in the coming months: (i) the possibility of monitoring in situ HP-SPS sintering; (ii) the extension of the (P,T) domain explored thanks to a new generation of adapted anvils; and (iii) adding plastic deformation during HP-SPS sintering.

### 4.1. In Situ X-ray Diffraction and Tomography during the HP-SPS Process

As mentioned in the introduction, the mechanisms involved in the SPS process are still under discussion today. The classical SPS sintering chamber remains a “black box” for researchers, which requires numerous blind tests to optimize the different parameters (pressure, heating and cooling rates, temperature and sintering time, etc.) for each new material. These multiple parameters of the SPS make any optimization very tedious, time-consuming and consuming in terms of energy and materials to be sintered. In situ X-ray diffraction has been used to follow the reaction mechanisms during current activated sintering, SHS (self-propagating high-temperature synthesis) [64] or the flash sintering of oxide powders, applying an electric field (100 V/cm) directly to the sample [65]. However, until now there has been no device for in situ monitoring of SPS processes. For this reason, we developed a compact very high-pressure SPS [52] on the base of a Paris–Edinburgh press, which is specially designed for in situ measurements on large facilities [22]. Indeed, the press and the pulsed current heating unit (described in section II-4) can be installed in one hour on a synchrotron beamline (Figure 17).

The pre-compacted powder is placed in a graphite mold and inserted into a gasket, following the principle described in Section 3.2 and in Figure 5A. The incident beam is aligned to pass in the gap between the two opposite anvils and through the gasket and the sample (Figure 17). To limit the X-ray absorption, the diameter of the sample is reduced to 2.5 mm or 1 mm depending on the nature of the sample and the desired pressure. The gasket made of amorphous boron-epoxy composite is X-ray transparent and serves as a thermal and electrical insulator as well as a pressure-transmitting medium [66]. We conducted time resolved X-ray diffraction experiments on an ID27 beamline at the European Synchrotron Radiation Facility (ESRF) and on a PSICHÉ beamline on a SOLEIL synchrotron. At ESRF, the wavelength of the high-energy X-ray beam was fixed to 0.2468 Å. The diffracted beam was collimated by a Sollers slits system filtering the diffraction reflections of the sample environment. The diffraction patterns of the sample were collected in transmission geometry using a MAR345 image plate scanner (X-Ray Research Company GmbH, Nodersted, Germany). At the SOLEIL synchrotron, energy dispersive X-ray diffraction (XRD) patterns were collected using a CAESAR detector [67] placed at a fixed angle of 8°. This angle has been precisely calibrated using an Au reference sample. The volume scanned with XRD was limited by the use of two sets of slits to a rhombus of 50 to 100 microns width, 500 to 1000 microns depth and 100 to 200 microns height. This technique probes the sample only allowing the collection of diffractograms without any reflection from its environment.

We followed the grain growth and the phase transition of an anatase TiO_2_ nanopowder during SPS sintering under 1.5 and 3.5 GPa [48] using in situ synchrotron X-ray diffraction. During heating to 725 °C at a rate of 8 °C/min, X-ray diffraction patterns of the sample were collected with a rate of one pattern per minute (Figure 18). In situ XRD highlights that pressure lowers the anatase-rutile transition temperature from 315 to 295 °C when pressure increases from 1.5 to 3.5 GPa, respectively. Indeed, at moderate pressure, the anatase to rutile phase transition usually occurs within 600–850 °C with a volume reduction of 9%. The application of high pressure during SPS sintering favors this transition.

In situ XRD gives access to the kinetics of the phase transformation. Using the Rietveld deconvolution method, the average lattice distortion and crystallite size were determined from the width of the Gaussian and Lorentzian function, respectively. Before the anatase-rutile transition, at around 300 °C, the anatase crystallite size increases from 15 to 25 nm. Then, the conversion anatase to rutile induces a jump in the crystallite size to 40 nm. This points to the formation of rutile grains mainly occurring by the merger of two anatase particles leading to a rapid grain growth at the phase transition. This study also shows that the application of high pressure slows down the growth of rutile crystallites. The growth rate is halved at 3.5 GPa compared to 1.5 GPa.

At SOLEIL, we studied the stability of the diamond phase during the SPS sintering of nanodiamond powders. Several batches of powders with different grain sizes (3, 5, 15 and 30 nm), from different synthesis routes were investigated. The in situ XRD experiment allowed us to find in a few experiments the minimum pressure required to preserve the diamond phase up to the high temperature needed for sintering. During heating up to 2000 °C at a rate of 200 °C/min, X-ray diffraction patterns of the sample were collected with a rate of six patterns per minute (Figure 19). Depending on the nature of the powder, the application of a pressure of at least 5 GPa can delay the graphitization until 2000 °C (unpublished results: SOLEIL Synchrotron Experimental Report Proposal 20210555). The optimum pressure for SPS sintering of a 15 nm diamond nanopowder was found to be 5 GPa. At this pressure, the graphitization of the diamond initiates at 2000 °C. Thanks to these results, we were able to sinter this powder at 5 GPa and 1900 °C, ex situ in the laboratory. This new in situ technique is therefore a major asset for optimizing the sintering conditions of strategic materials, such as nanodiamond compacts.

These first experiments have shown the relevance of this in situ technique for the study of the transformation kinetics of materials as a function of the characteristics of the current and the pressure applied. This type of real-time analysis is not possible on any other SPS device in the world.

Extending the use of this compact very high-pressure SPS to ultrafast synchrotron X-ray tomography will allow in situ following of grain shape change during coalescence, the evolution of the porosity during the sintering process, or the growth of a reactive interface in case of composites, multi-layer materials or reactive sintering. Fast tomography measurements at high temperature and high pressure, using a modified Paris–Edinburgh press, called Ultra-fast Tomography Paris–Edinburgh cell (UToPEc) [67,68], is already available on a PSICHE beamline at Synchrotron SOLEIL [67,68,69,70]. The UToPEc is a panoramic (i.e., 165° angular aperture) press optimized for fast tomography that can access 10 GPa and 2000 °C. It is installed on a high-speed rotation stage (up to 360°/s) and allows the acquisition of a full X-ray computed tomography image with micron spatial resolution within a second. The sample cell assembly (Section 3.2 and Figure 5A) used for in situ XRD are compatible with this tomography set-up. The adaptation of the pulsed current heating unit on the UToPEc press will mark a new breakthrough for the in situ monitoring of the SPS-HP process.

### 4.2. Extension of the (P,T) Domain Explored

#### 4.2.1. Higher Pressure

Increasing the maximum pressure in HP-SPS experiments can only be achieved by playing on three parameters: sample size, hydraulic press capacity or primary anvil materials.

Firstly, the performance can obviously be increased by a reduction of the sample volume. Actually, these HP-SPS devices are based on the principle of intensification: the hydraulic press allows a relatively moderate primary pressure to be applied on the base of the piston and the force generated is transmitted over a smaller area between the anvils, which leads to the intensification of the pressure obtained on the sample. Decreasing this surface area (and therefore the gasket and the size of the sample) ipso facto leads to an increase in the maximum pressure reached. Nevertheless, since the gasket deforms plastically, there is no exact proportionality relationship between the size and the maximum pressure. Thus, for example, in the field of the Paris–Edinburgh apparatus mentioned in 3.4, reducing the sample size from 8 mm to 1 mm (in dedicated sample assembly) makes it possible to increase the pressure by a factor of two with CW anvils. However, such an optimization of the maximum pressure by reducing the sample is not commonly pursued in the case of sintering experiments since generally large sintered materials (at least of the order of 10 mm in diameter) are required in order to properly study their properties or use them in application areas.

The second option is to use the principle of intensification in the other direction, i.e., no longer decrease the sample size but increase the capacity of the press (and therefore its size) in order to increase the primary pressure applied to the piston and therefore the force transmitted to the gasket with unchanged size. In the field of high-pressure technology, this optimization has already been successfully carried out. For example, Irifune et al. [71], thanks to a gigantic dedicated press with a record capacity of 60 MN, have made it possible to produce sintered nanodiamonds with dimensions to 1 cm in both diameter and length at 16 GPa and 2500 °C. Coupled with a pulsed DC generator, this kind of high-pressure equipment could therefore theoretically allow sintering of samples of 10 mm in diameter up to 16 GPa and 2500 °C. Nevertheless, the size and weight of this kind of press are considerable, which somewhat limits their development in academic laboratories.

The last and most effective option for increasing the maximum pressure is to change the hard material of the anvil. All HP-SPS technologies are currently equipped with WC anvils. Tungsten carbide has a compressive strength of about 6.5 GPa. Supported WC anvils are therefore, in practice, limited to ~10 GPa, irrespective of the details of the anvil design. The replacement of WC anvils with sintered diamond anvils allows in all high-pressure technologies to gain an important factor on maximum pressure. Thus, for example, in the field of the Paris–Edinburgh press mentioned in 3.4, the use of sintered diamond anvils makes it possible to increase the maximum pressure by a factor of 1.5 compared to WC anvils with the same design [72]. The problem most often encountered is allowing the passage of electric current in its sintered diamond anvils since they are weakly conductive of electricity. However, recent developments in progress (with electrodes drilled directly from the anvil) will soon allow this difficulty to be overcome and will quickly offer an extension of the pressure range while maintaining the same accessible temperature range. These devices will be detailed in a dedicated paper. Finally, it should be added, since pressure and sample volume are two complementary parameters, that one could imagine using these new sintered diamond anvils to increase the sample volume at constant maximum pressure.

#### 4.2.2. Higher Temperature

A number of changes are required to extend the temperature limits and these changes are dependent on the target pressure range. At moderate pressure (up to 8 GPa), new sample chambers adapted to the various high-pressure devices used in the HP-SPS presses mentioned above could be tested. For example, the HP-SPS device working with the cubic press [54] could benefit from recent experimental optimizations of the high-pressure cubic cell assembly where a record temperature value of ~3700 °C has been obtained [73]. Similar developments have been made on belt-type [74] or toroidal-type [75] presses which could be very useful for extending temperature range in HP-SPS technologies based on these pressure generators. At higher pressure (>8 GPa), the central problem comes from the graphite (furnace) since heating is limited by the graphite–diamond conversion of the heater at high pressure and temperature. Some other high resistivity compounds (such as boron-doped diamond, TiC or LaCrO_3_), which have been already used in many high-pressure technologies, could increase the temperature range without a major change in the geometry of the sample chamber [74,76,77]. Nevertheless, the temperature stability using such heaters needs to be carefully investigated. Furthermore, an extension of the temperature range, if it will allow the sintering of very refractory compounds to be studied, will necessarily increase the temperature gradients inside the sample. Therefore, studies should be conducted to estimate the thermal gradients and optimize the sample chamber’s design using finite element calculations as is performed for classic SPS assemblies [78,79,80].

### 4.3. Adding Plastic Deformation during HP-SPS Sintering

It is widely accepted that the application of severe plastic deformation (SPD), in which the sample undergoes measurable plastic stress under high pressure, leads to exceptional grain refinement in metallic materials and, hence, is a promising tool to obtain a wide range of nanocrystalline metals and alloys with exceptional mechanical properties [81,82,83]. Several publications over the past few decades have shown that the SPD method can also control phase transformations and lead to grain refinement not only in metals, but also in various non-metallic materials [84,85,86,87,88,89]. These numerous structural changes offer new possibilities to tune and obtain major improvements in the physical, mechanical, chemical and functional properties of materials.

This SPD methodology can be advantageously combined with HP-SPS technology. Indeed, Muche et al. [90] showed that the effect of pressure on grain growth in SPS, while effective in drastically reducing it, is rarely 100% effective. Indeed, to eliminate unavoidable small residual porosity at high pressure, there is always limited grain growth to allow complete densification. The SPD technique can suppress this effect by reactivating densification through mechanical grain sliding, without the need for thermal grain growth to eliminate residual porosities. Thus, it would be theoretically possible by the combined effect of SPD and HP-SPS to more easily obtain completely dense nanostructured materials.

Currently two kinds of devices effectively combine SPD and HP-SPS in the literature. Both are especially attractive for future research in this exciting new field. The first one, named DP-SPS (for deformable punch spark plasma sintering) [90] or SPT (for spark plasma texturing) [91] uses a special HP-SPS sample chamber so that uniaxial pressure deforms the sample in the direction perpendicular to applied pressure (Figure 20). SPT allows operation at moderate pressure up to 500 MPa, and DP-SPS at high pressure up to 2 GPa. The principle is to plastically deform the sample by allowing it to deform freely (for SPT) or almost freely (for DP-SPS, as surrounded by a soft material, like graphite) in the plane perpendicular to the uniaxial pressure applied. Nevertheless, the deformation in these systems is difficult to quantify, as it is very dependent on various parameters.

The second kind of SPD-HP-SPS device is based on the principle of high-pressure torsion (HPT) in which the sample undergoes measurable plastic strain by applying an opposite torsion on the two rotating anvils under high pressure. A first basic concept idea was put forward in 1999 in a US patent [92]. However, it is unclear whether the system was actually built and put into operation as there have been no further publications by the team involved since. Moreover, in this device, the shear force was limited at a first stage of the SPS processing, where the pressure is about only 5–50 MPa. Hence, it cannot be considered as true SPD-HP-SPS. In contrast, we recently developed such an HPT device by integrating a torsional module (RoToPEc module) into our HP-SPS system [52] (Figure 21).

While the detailed working principle and description of this RoToPEc module can be found elsewhere [93], one can briefly mention that, in this system, the two opposed anvils compressing the sample can rotate independently under high load with no limitation in the rotation angle, through two sets of gear reducers and thrust bearings (Figure 21). Stepper motors and encoders, with an angular resolution of 0.02°, allow monitoring the accurate rotation of the anvils and hence the deformation of the sample. The press is heavier than the conventional Paris–Edinburgh press at 197 kg because of its additional rotational components. Nonetheless, it can be easily disassembled and transported for performing in situ experiments (diffraction X or tomography X). This SPD-HP-SPS device allows operation at high pressure up to 7 GPa and is currently widely used for sintering various materials.

As a proof of concept, using DP-SPS, Muche et al. obtained fully dense nanocrystalline magnesium aluminate (MgAl_2_O_4_) ceramics with exceptional Vickers hardness (28.4 GPa, surpassing the hardness of sapphire) and transparency [90]. These remarkable properties are linked to the total absence of pores and to the very small grain size (7 nm) which leads to an extensive network of grain boundaries. This first study shows that coupling deformation and HP-SPS may be the best way in the future to achieve porosity-free nanoceramics and reveal their potential.

## 5. Conclusions

By combining SPS and very high-pressure technologies, the pressure limit of the SPS process has recently been extended to 10 GPa, opening up new opportunities in the development of advanced materials elaboration, especially for hard-to-sinter and heat-sensitive materials and nanomaterials. The HP-SPS field is currently a rapidly expanding area with various types of devices, based on optimized high-pressure technological concepts.

In this review, we have exhaustively described and given the main characteristics (pressure and temperature ranges, sample volume, metrology and working principle for compression) of the currently existing HP-SPS devices operational above 500 MPa. Each technology has its own particular features, such as (P,T) range, uniaxial or quasi-hydrostatic compression, the possibility or not of performing in situ studies, etc.

In future, these devices can be used in a complementary way to optimize a material process. They are also perpetually undergoing developments and improvements. Hence, we provided some prospective routes to extend their use to ultrafast tomography, to widen their (P,T) domain and to combine them with severe plastic deformation.

Contrary to the common belief, high-pressure technology is adaptable to large-volume industrial production, as illustrated by the markets of high-pressure synthetic diamonds or cutting tools with other ultra-hard materials (e.g., cubic boron nitride). Thus, the HP-SPS domain is only at the beginning of its promising academic and industrial history.

## Figures and Tables

**Figure 1 materials-16-00997-f001:**
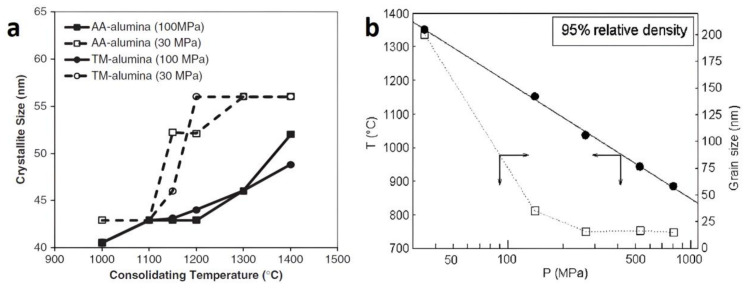
(**a**) Dependence of crystallite size on SPS with temperature and applied pressure for two different commercial powders of alumina (TM-alumina and AA-alumina) (from [26]). (**b**) Relationship between the pressure required and the dwell temperature (5 min) to obtain samples with a relative density of 95% in the case of nanometric zirconia (8% YO_1.5_). The full circles correspond to the temperature of sintering and empty squares to the grain size obtained (from [25]).

**Figure 2 materials-16-00997-f002:**
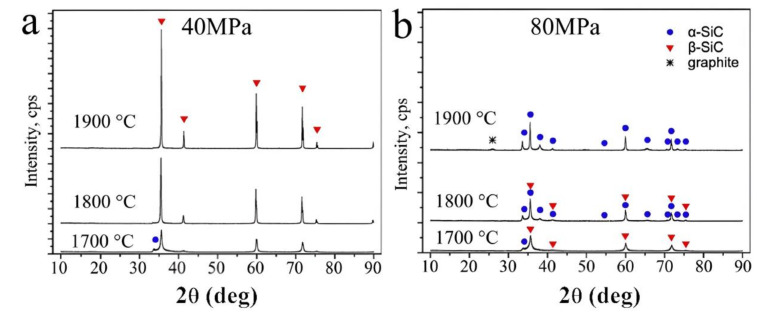
X-ray diffraction patterns (CuKα) of SPS-processed silicon carbide at (**a**) 40 and (**b**) 80 MPa. The blue symbols represent the diffraction peaks of α-SiC and the red symbols the diffraction peaks of β-SiC [35].

**Figure 3 materials-16-00997-f003:**
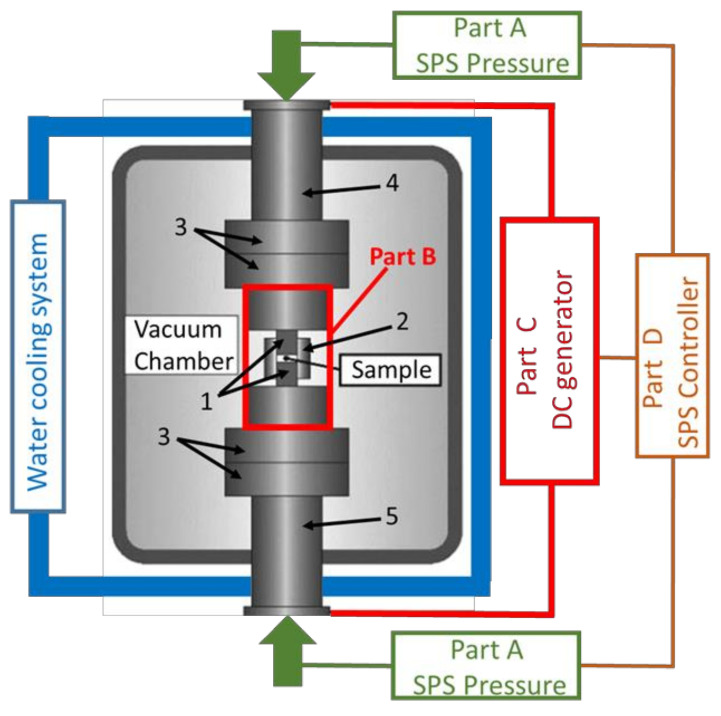
Schematic of an SPS apparatus. (1) Graphite punches. (2) Cylindrical die (usually in graphite). (3) Graphite spacers. (4) Upper water-cooled steel cylinder. (5) Lower water-cooled steel cylinder. Adapted from [25,42].

**Figure 4 materials-16-00997-f004:**
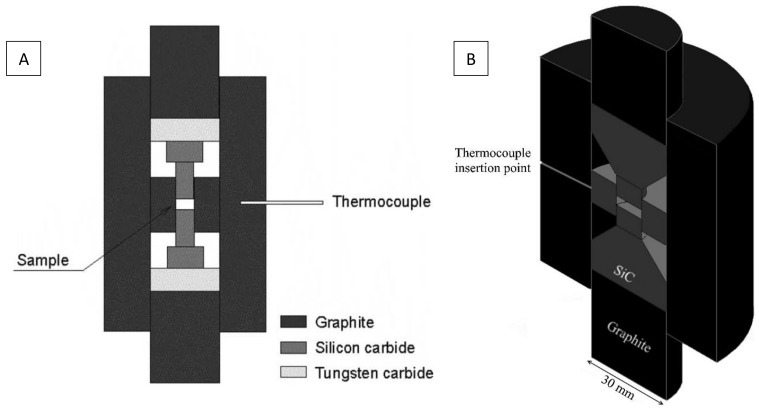
Schematics of HP-SPS devices reaching 1 GPa on 5 mm (**A**) [25] and 10 mm (**B**) [50] diameter samples.

**Figure 5 materials-16-00997-f005:**
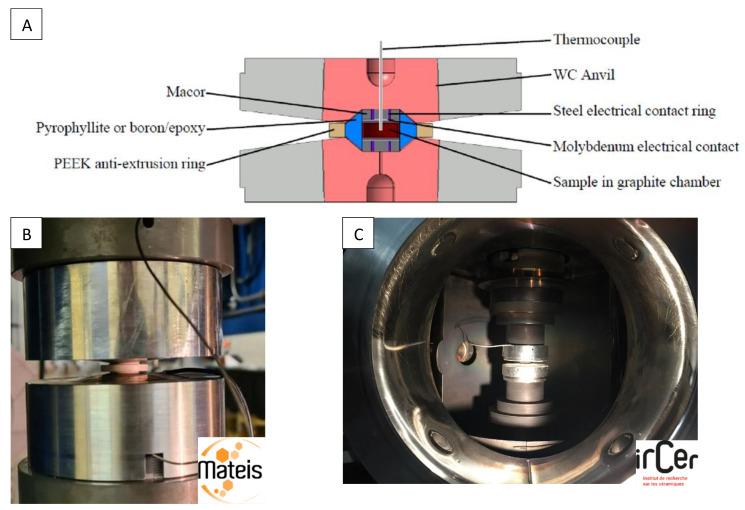
(**A**) Section view of the sample cell assembly between two WC anvils (**B**) The Paris–Edinburgh HP-SPS module mounted in the SPS-HPD 25 (FCT) chamber and (**C**) in the Dr. Sinter SPS 825 chamber, working up to 2 GPa. Experiments were performed in two different laboratories: MATEIS, Lyon (France) and IRCER, Limoges (France).

**Figure 6 materials-16-00997-f006:**
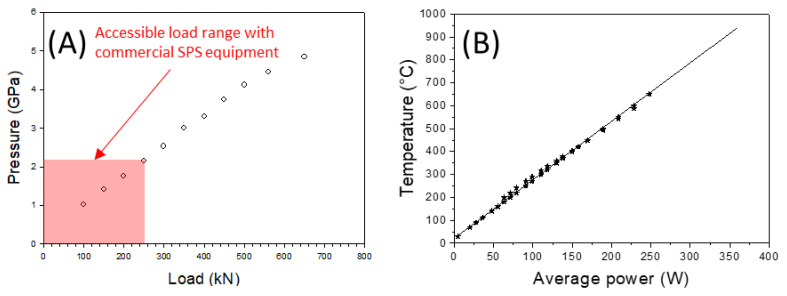
Calibrations of the Paris–Edinburgh HP-SPS cell: (**A**) Calibration curve of the pressure performed from in situ neutron diffraction of a NaCl sample using its equation of state. The orange area indicates the pressure range accessible with the Paris–Edinburgh module mounted on standard SPS equipment (maximum load 250 kN) (**B**) Calibration curve of the temperature, as a function of the average power of the pulsed current, performed by placing a K-type thermocouple in the center of an alumina sample.

**Figure 7 materials-16-00997-f007:**
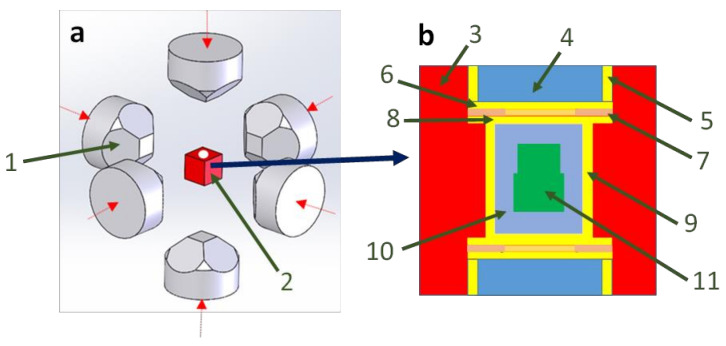
(**a**) Schematic of cubic press assembly showing the six opposite WC anvils and the cube assembly in the middle; (**b**) Enlargement of the associated cubic reaction chamber. (1) WC anvils. (2) Cubic reaction chamber. (3) Gasket material. (4) “Synthetic gasket material”. (5) Steel electrode. (6) First metallic disk. (7) Baffle. (8) Second metallic disk. (9) Graphite heater. (10) Salt capsule. (11) Metallic capsule where the sample is placed. The red arrows indicate compression axes. Adapted from [22,54].

**Figure 8 materials-16-00997-f008:**
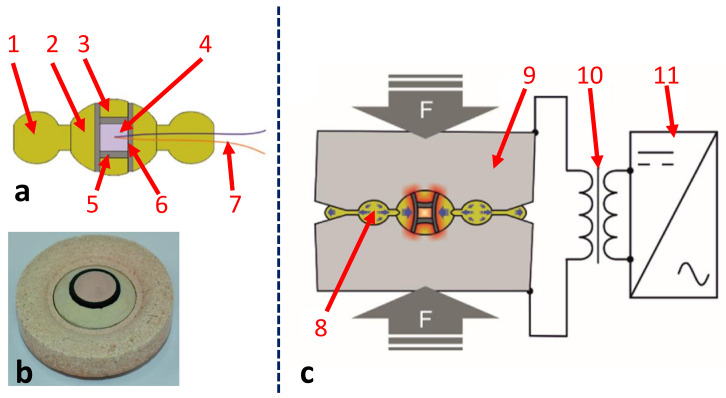
HP-SPS device at the Institute of Advanced Manufacturing Technology (at Krakow) in Poland. Cross section (**a**,**c**) and photography (**b**) of the high-pressure reaction chamber and HP-SPS device schematic, where: 1—ceramic gasket (outer part); 2—ceramic gasket (inner part); 3—ceramic disc; 4—sample; 5—graphite disc; 6—graphite tube; 7—thermocouple (used only for temperature calibration). Quasi-isostatic compression of the preliminary consolidated powders is achieved as a result of plastic deformation of the gasket material (8) between anvils (9); electrical heating during HP-SPS process is provided by transformer (10), together with inverter (11), thus providing 1 kHz direct pulsed current. Adapted from [57,58].

**Figure 9 materials-16-00997-f009:**
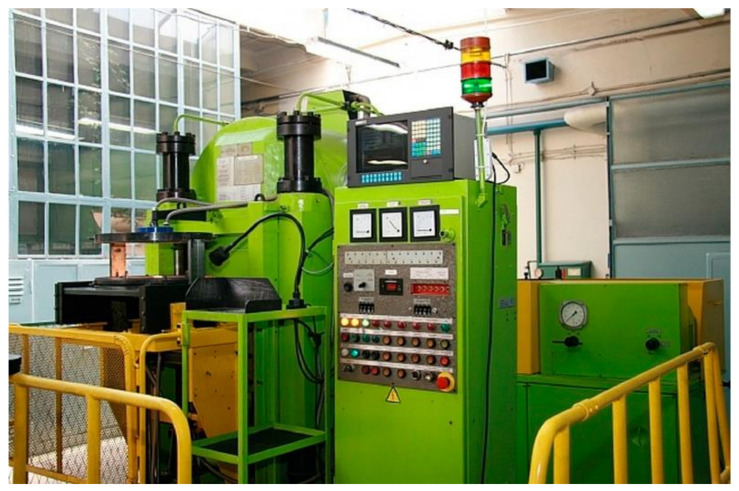
HP-SPS device at the Institute of Advanced Manufacturing Technology (at Krakow) in Poland (from [57]).

**Figure 10 materials-16-00997-f010:**
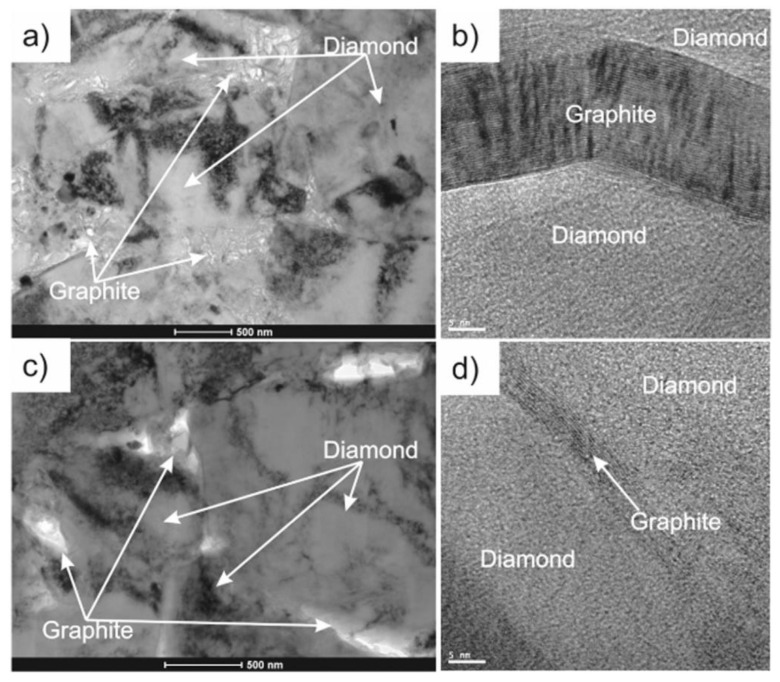
Microscopic images of the samples with 90 wt% diamond +10 wt% (Ti+ 2B). TEM image (**a**) and HREM image (**b**) of diamond composite sintered with classical HP–HT technique. TEM image (**c**) and HREM image (**d**) of diamond composite sintered with HP-SPS technique (from [58]).

**Figure 11 materials-16-00997-f011:**
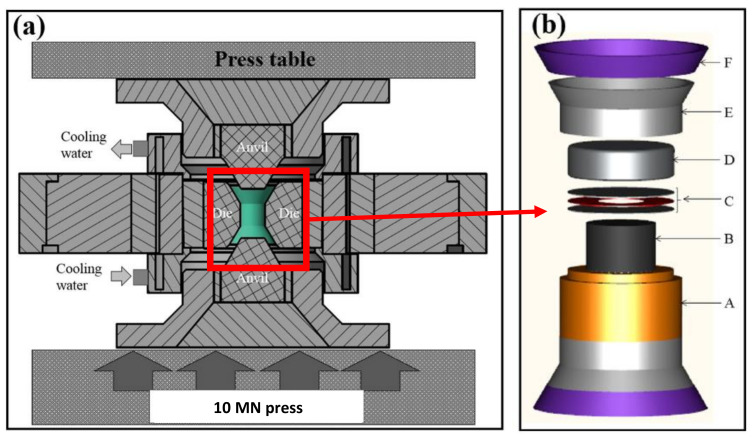
(**a**) HP-SPS belt apparatus showing the high-pressure reaction chamber (in green), where the sample is located, and the external support (the “belt”), used to contain the pressure. (**b**). High-pressure reaction chamber: A: Fired pyrophyllite tube; B: Graphite heater; C: Molybdenum discs sandwiching mica ring; D: Steel cover filled with fired pyrophyllite pellet; E: Crude pyrophyllite gasket; F: Polymer gasket (adapted from [59]).

**Figure 12 materials-16-00997-f012:**
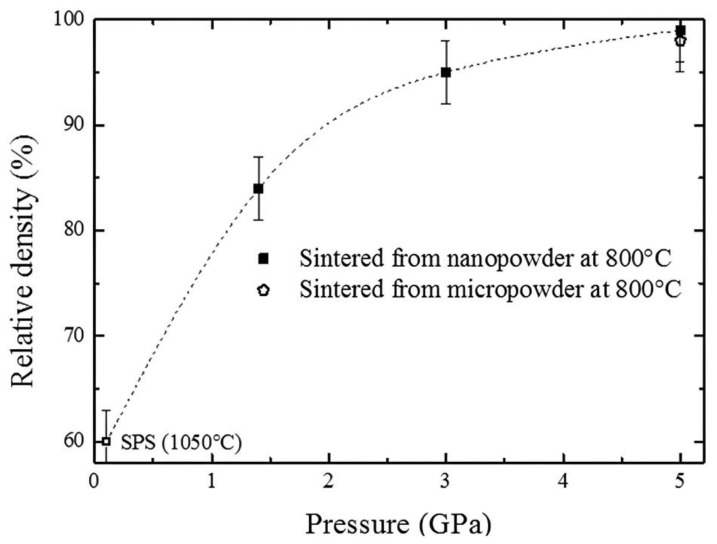
Density evolution with pressure in recovered α-Al_2_O_3_ ceramics sintered at 800 °C from γ-Al_2_O_3_ with nano or micro-sized grains. The first symbol at lowest pressure corresponding to conventional SPS (100 MPa). (Figure adapted from [61]).

**Figure 13 materials-16-00997-f013:**
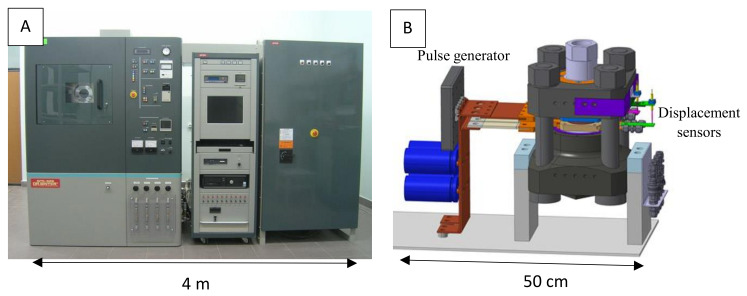
An SPS apparatus commercialized by FUJI Electronic Industrial Co., Ltd. (Tsurugashima, Japan) (**A**) and the tabletop HP-SPS device (**B**).

**Figure 14 materials-16-00997-f014:**
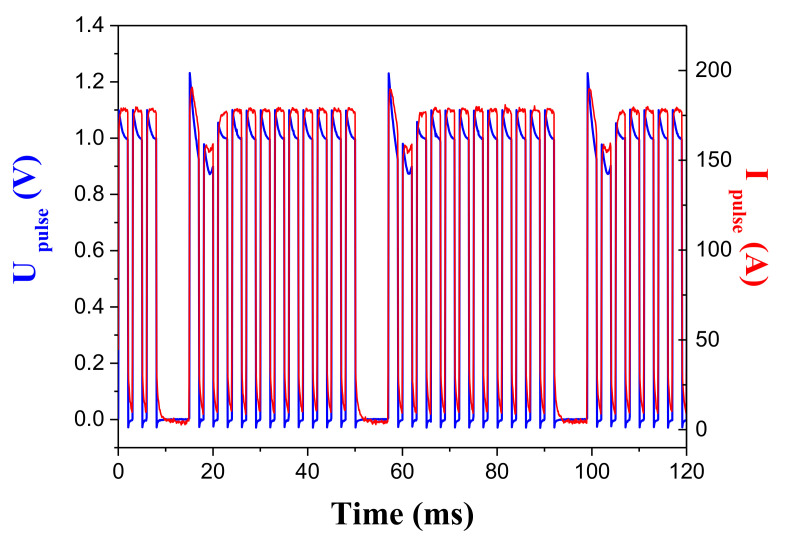
SPS pulse pattern, with an On:Off setting of 12:2, delivered using the homemade pulse generator to the tabletop HP-SPS.

**Figure 15 materials-16-00997-f015:**
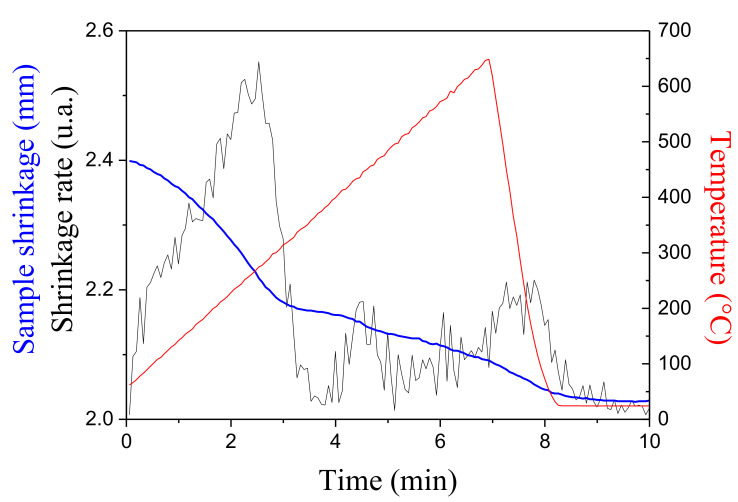
Sintering curve of an anatase-TiO_2_ nanopowder recorded under 1 GPa upon heating up to 650 °C.

**Figure 16 materials-16-00997-f016:**
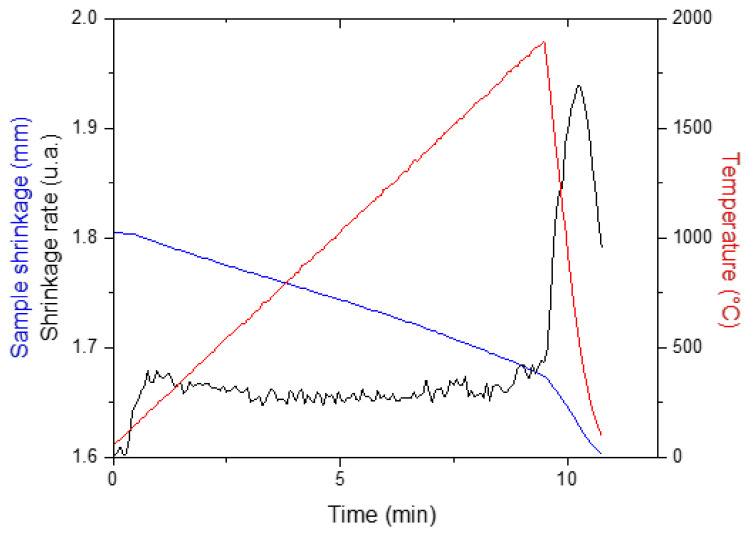
Sintering curve of a diamond nanopowder recorded under 5 GPa upon heating up to 1900 °C.

**Figure 17 materials-16-00997-f017:**
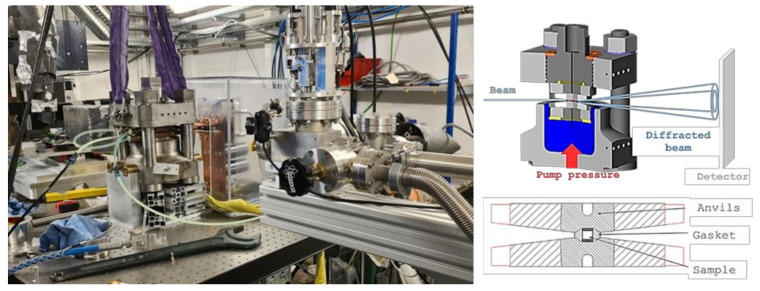
The compact very high-pressure SPS installed on PSICHÉ beamline at SOLEIL, French synchrotron. Scheme of the press set-up for collecting diffraction patterns—Details of the sample environment.

**Figure 18 materials-16-00997-f018:**
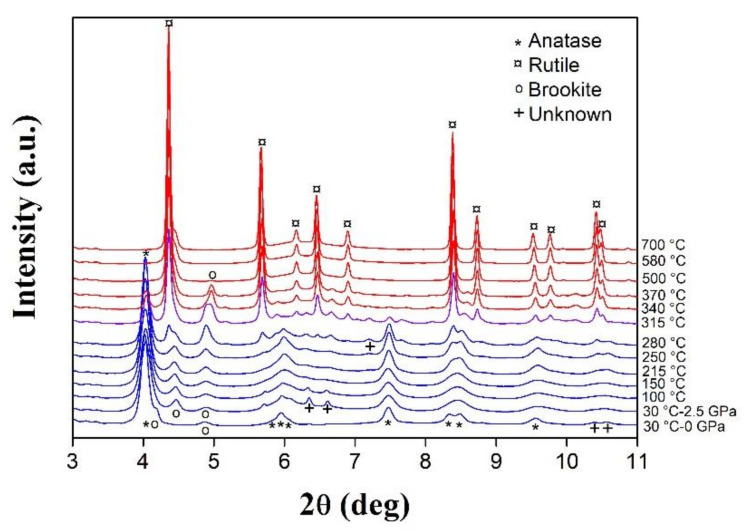
In situ X-ray diffractograms collected on heating during SPS treatment of a 15 nm TiO_2_ nanopowder at 3.5 GPa (ID27 beamline, ESRF, λ = 0.2468 Å).

**Figure 19 materials-16-00997-f019:**
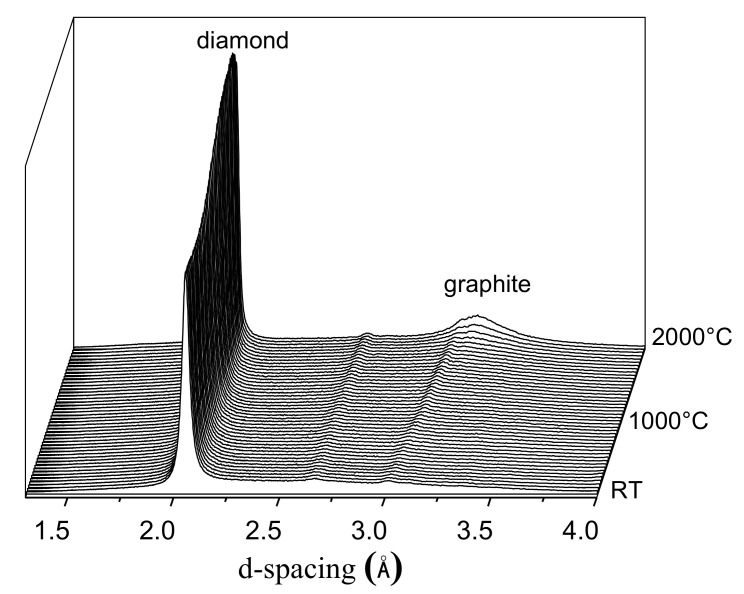
In situ X-ray diffractograms collected on heating (200 °C/min) during HP-SPS treatment of a diamond nanopowder (PSICHE beamline, SOLEIL, in energy dispersion at a fixed angle).

**Figure 20 materials-16-00997-f020:**
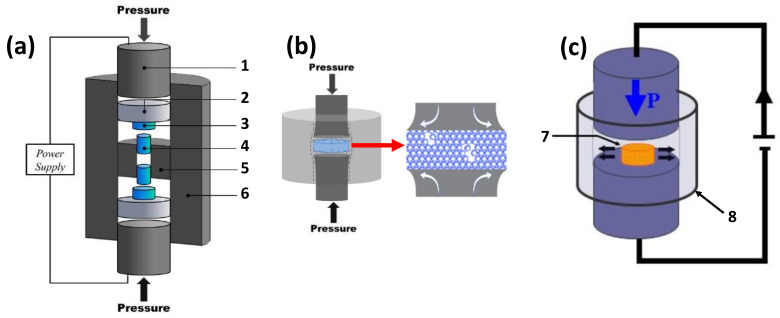
(**a**) DP-SPS configuration. 1. Graphite punch; 2. Silicon carbide space; 3. Tungsten carbide spacer; 4. Tungsten carbide punch; 5. Internal graphite die; 6. Graphite die. (**b**) Schematic of the sample being compressed by tungsten carbide punches inside the graphite die, during sintering. (**c**) Spark plasma texturing (SPT) configuration. 7. Pre-sintered sample; 8. Mold. Adapted from [90,91].

**Figure 21 materials-16-00997-f021:**
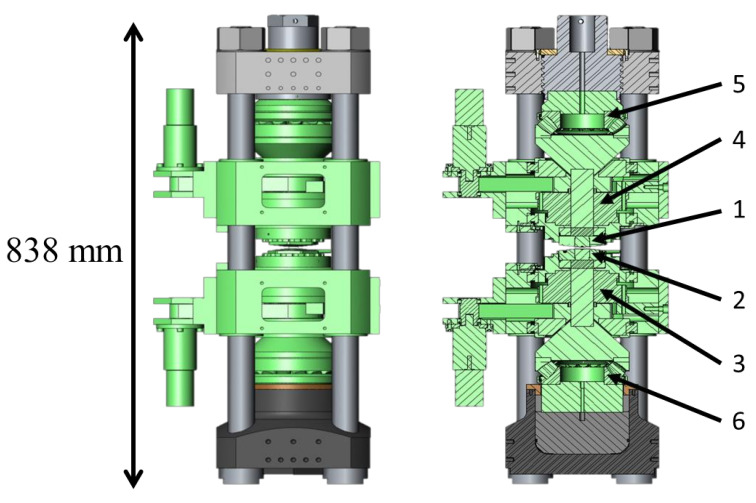
3D scheme and cross-section of the RoToPEc module which can be installed on HP-SPS device [52]: (1) rotating upper anvil, (2) rotating lower anvil, (3) lower gear reducer, (4) upper gear reducer, (5) upper thrust bearings, (6) lower thrust bearings.

## Data Availability

No new data were created or analyzed in this study. Data sharing is not applicable to this article.

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
