# Peer review of "Recent Developments of High-Pressure Spark Plasma Sintering: An Overview of Current Applications, Challenges and Future Directions"

_materials, 2023, doi:10.3390/ma16030997_

Round 1

Reviewer 1 Report

The article "Recent High-Pressure Spark Plasma Sintering Developments: An Overview of Current Applications, Challenges, and Future Directions" is new and interesting enough to warrant publication.

The literature review about "High-Pressure Spark Plasma Sintering" significantly contributes to the conceptual, methodological, and thematic development. Since there is no general agreement on the details of the technique, an unequivocal definition of SPS and its associated procedures cannot be defined.

However, a bibliometric analysis to visualize trends and patterns in the relevant literature (thus clarifying the authors' scientific contribution and citation potential) is a good option for creating a literature review (manuscript).

In my opinion, it also lacks the most important conclusions. The authors could present global and specific provisions in relation to the objectives of their research.

Reviewer 2 Report

The Review dealing with High-Pressure Spark Plasma Sintering is well written and composed. The individual chapters are logic and giving relevant information.

Since it is review I would appreciate some schematic explanation of facts e.g. explaining advantages and disadvantages of individual techniques.

In the text should be unified:  units and expressions of different properties.

In the Fig 2  x -axis should be labeled in symbols theta.

Review should have conclusion at least as future outlook.

Reviewer 3 Report

General

This is a review paper of mostly the hardware development in the SPS field. It also presents the development of the field and the process associated with it.

This manuscript served well to fill a gap for someone who is new in the field and provide a blueprint for how to use the SPS technique to synthesize materials, such as crystalline size as a function of sintering temperature and applied pressure.

This manuscript reviewed the SPS subject area and is beneficial to the general scientific community. It educates a wide audience in an advanced material processing technique.

Methodology

While the manuscript mainly focused on the development of the hardware, it can be improved by adding more information on the basic science area, such as process parameters on atomic diffusion and other kinetic mechanisms. More explanations of the parameters will be valuable, such as how the SPS pulse pattern is optimized. A short description of these parameters will be useful for someone new to the field. Since this is a review paper, there is no original methodology or controls presented.

Conclusions

This is a review manuscript. Hence, no conclusions or arguments from the original experiments. However, the authors may add a conclusion to summarize the topics covered in this manuscript.

Additional Comments

·        Check the format consistency, such as “Figure” vs “figure”

·        Line 207 – Is there a reason to use 12-2 pulse? (Also mentioned in Figure 14) Maybe some explanations for the reason behind this 12-2 ratio?

·        Line 224 – “many research groups”, consider adding references

·        Line 325 – Niobium, lowercase for this word?

Reviewer 4 Report

The topic of the work is important, relevant, and promising. The authors have collected large-scale material on the topic. But there are some remarks on the organization of the text and, especially, on the analysis and interpretation of facts.

P1 L31-32 “The Spark Plasma Sintering (SPS) is a sintering technique under moderate pressure (up to 150 MPa) and at high temperature (up to 2000°C)”.

For a sufficiently large group of UHTCs, this assertion is incorrect. Here are several cases of SPSing at temperatures above 2000 °C (for example only, not a complete list):

Solid solution formation and sintering behavior of TaC–HfC ceramics made from commercial TaC and HfC powders prepared using spark plasma sintering (SPS) at temperatures up to 2450 °C was investigated [https://doi.org/10.1016/j.jeurceramsoc.2016.02.009].

TiC–ZrC composites were consolidated from TiC and ZrC powders by spark plasma sintering (SPS) at 1500–2200 °C [https://doi.org/10.1016/j.ceramint.2015.02.019].

Figures 2 show the relative density and grain size of ZrB2 samples obtained by CSPS (Figures (a)) and FSPS Figures (b)) sintered at different temperatures and for different discharge times. In agreement with the displacement rate shown in Figure 1 (a), densification in FSPS occurred between 15 and 35 seconds, during which the density increased from 75.9 to 95.0%. As shown in Figure 1(c), the peak temperature was 2198 áµ’C after 35 s of discharging. In CSPS when the temperature was increased from 2000 to 2100áµ’C the grain size sharply increased from 6 to 18.2 µm. Similarly, in the case of FSPS the grain growth was significant only when the processing temperature was above 2000 áµ’C; the average grain size increased from 2.6 to 11.8 µm for the samples FSPSed for 25 and 35 s [https://doi.org/10.1111/jace.13109].

P1-P2 It may be worthwhile to define the terms in the introduction. What is the main sign of a high-pressure SPS - the use of stronger molds instead of graphite (then the value of 150 MPa is quite acceptable), or the principle of quasi-isostatic pressing instead of uniaxial (then the borderline is not so clear)? Or it is not necessary to give exact definitions at all but to consider HP SPS as a gradual development of conventional SPS technology.

P3 L128-129 “…the time required for atomic diffusion and mass transfer in the process is more limited when pressure is applied…” Somewhat ambiguous wording: limited time = mass transfer takes less time. But it can also be understood as a lack of time for mass transfer.

Sections 3.2, 3.4, and 4.1 makes the reader question whether he is looking at a review or an experimental work (… we adapted…). Perhaps it is worth sticking to the style of presentation adopted at the beginning. Otherwise, it would be a good idea to warn readers in the introduction that the work is a combination of a literature review with a description of the authors' own achievements, published for the first time.

P12 L400 – P13 L408 Unfortunately, there is only a listing of the facts from [58], but not their analysis (like in most other cases). It would be interesting to read the author's interpretation of this difference between HP-HT and SPS-HP.

The work lacks conclusions that would sum up and explain the relationship between all the facts presented. Without conclusions, the work looks incomplete. A weak attempt at generalization and prediction was made in Section 4. Prospects for technological breakthroughs, but this is clearly not enough.
